# Activation of the Complement System in Patients with Cancer Cachexia

**DOI:** 10.3390/cancers13225767

**Published:** 2021-11-17

**Authors:** Min Deng, Rianne D. W. Vaes, Annemarie A. J. H. M. van Bijnen, Steven W. M. Olde Damink, Sander S. Rensen

**Affiliations:** 1Department of Surgery, NUTRIM School of Nutrition and Translational Research in Metabolism, Maastricht University, 6229 ER Maastricht, The Netherlands; min.deng@maastrichtuniversity.nl (M.D.); r.vaes@maastrichtuniversity.nl (R.D.W.V.); a.vanbijnen@maastrichtuniversity.nl (A.A.J.H.M.v.B.); steven.oldedamink@maastrichtuniversity.nl (S.W.M.O.D.); 2Department of General, Visceral and Transplantation Surgery, RWTH Aachen University, 52074 Aachen, Germany

**Keywords:** cancer cachexia, complement system activation, inflammation

## Abstract

**Simple Summary:**

Patients with cancer often suffer from severe weight loss as a result of the wasting of skeletal muscle and fat tissue. This has a strong negative impact on the patient’s prognosis and quality of life. Inflammation is thought to contribute to weight loss in cancer. To enable the future targeting of inflammation in patients with cancer who experience weight loss, we set out to characterize an important pro-inflammatory component of the immune system: the complement system. The blood levels of several elements of the complement system were analyzed in patients with and without weight loss and inflammation. We found that complement factors were activated specifically in patients with both weight loss and inflammation. Since complement inhibitory drugs are already on the market, these findings may open new opportunities for treating inflammation-mediated weight loss in patients with cancer.

**Abstract:**

Systemic inflammation is thought to underlie many of the metabolic manifestations of cachexia in cancer patients. The complement system is an important component of innate immunity that has been shown to contribute to metabolic inflammation. We hypothesized that systemic inflammation in patients with cancer cachexia was associated with complement activation. Systemic C3a levels were higher in cachectic patients with inflammation (*n* = 23, C-reactive protein (CRP) ≥ 10 mg/L) as compared to patients without inflammation (*n* = 26, CRP < 10 mg/L) or without cachexia (*n* = 13) (medians 102.4 (IQR 89.4–158.0) vs. 81.4 (IQR 47.9–124.0) vs. 61.6 (IQR 46.8–86.8) ng/mL, respectively, *p* = 0.0186). Accordingly, terminal complement complex (TCC) concentrations gradually increased in these patient groups (medians 2298 (IQR 2022–3058) vs. 1939 (IQR 1725–2311) vs. 1805 (IQR 1552–2569) mAU/mL, respectively, *p* = 0.0511). C3a and TCC concentrations were strongly correlated (r_s_ = 0.468, *p* = 0.0005). Although concentrations of C1q and mannose-binding lectin did not differ between groups, C1q levels were correlated with both C3a and TCC concentrations (r_s_ = 0.394, *p* = 0.0042 and r_s_ = 0.300, *p* = 0.0188, respectively). In conclusion, systemic inflammation in patients with cancer cachexia is associated with the activation of key effector complement factors. The correlations between C1q and C3a/TCC suggest that the classical complement pathway could play a role in complement activation in patients with pancreatic cancer.

## 1. Introduction

Cachexia is a severe complication of many types of cancer, with a particularly high prevalence in pancreatic cancer [1]. It is a multifactorial metabolic syndrome that has a substantial detrimental impact on the survival and quality of life of patients [2]. Cancer cachexia is characterized by loss of body weight, which negatively impacts both adipose tissue and skeletal muscle mass [3].

Although its precise underlying mechanisms remain unclear, many lines of evidence point towards an important role of the pro-inflammatory factors released by tumor cells, immune cells, or metabolic target tissues in the pathophysiology of cancer cachexia. In particular, inflammation has been shown to be associated with both muscle wasting and fat depletion in patients with cancer cachexia [4,5]. Interactions between the tumor and host cells leading to systemic inflammation result in an acute-phase response characterized by the production of C-reactive protein (CRP) in the liver and increases in circulating proinflammatory cytokines [6,7]. In recent years, research has mainly focused on the role of pro-inflammatory cytokines, such as interleukin-6 (IL-6), interleukin-1 (IL-1), tumor necrosis factor alpha (TNF-α), and interferon gamma (IFN-γ), and confirmed their involvement in the pathogenesis of cachexia [8,9,10,11].

Despite these studies, the characterization of the systemic inflammatory response in patients with cancer cachexia is still incomplete. For example, there is no literature on the potential involvement of the complement system, a cornerstone of innate immunity that plays a key role in cancer-associated cachexia. The complement system comprises over 50 circulating, cell surface-bound or intracellular proteins that collectively promote the recognition and removal of both infectious agents and apoptotic, necrotic, or malignant cells [12,13], the latter of which may be highly relevant in the context of tissue wasting in cancer cachexia. Complement molecules may be expressed by activated immune cells including macrophages, dendritic cells, natural killer cells, B cells, and T cells. They can also be produced by other cell types involved in the metabolic aberrations seen in cachexia, such as hepatocytes and adipocytes [14,15]. Various studies have reported elevated levels of complement factors in cancers with a high prevalence of cachexia, including colorectal cancer and lung cancer [16], further supporting a role for complement system activation in cachexia-associated inflammation.

There are three different ways to activate the complement system, referred to as the classical pathway, the lectin pathway, and the alternative pathway. The classical pathway of complement activation is initiated by binding of the C1q molecule to antigen-antibody complexes, CRP, or apoptotic cells [17]. Lectin pathway activation is dependent on the recognition of aberrant carbohydrate patterns by the mannose-binding lectin (MBL) protein, collectins, or ficolins [18]. The activation of the alternative pathway can be initiated by the spontaneous activation of the central complement C3 molecule on ‘activating surfaces’; this pathway also serves to amplify complement activation as a result of classical or lectin complement pathway activation [19].

Although the three pathways of complement activation differ with respect to their method of target recognition, they all converge at the level of the C3 molecule, which is activated through cleavage by C3 convertases generated by the respective pathways. Upon its activation, several effector complement molecules with different functions are generated: the opsonin C3b, the anaphylatoxins C3a and C5a, and the membrane attack complex (MAC). The deposition of the C3b opsonin on complement-activating surfaces enhances the phagocytosis potential of macrophages and promotes their secretion of pro-inflammatory cytokines [20]. Anaphylatoxins, such as C3a, are potent chemo-attractants that recruit monocytes and granulocytes to the site of activation and can stimulate the production of additional inflammatory mediators. C3a has been shown to enhance pro-inflammatory cytokine secretion by various immune cells [21]. For example, Elvington et al. reported that C3a increased IL-6 production by CD4+ T cells, contributing to lipolysis [22]. C3a also stimulates the production of cachexia-associated inflammatory mediators such as TNF-α and IL-1 [23,24]. The assembly of the MAC is the ultimate consequence of complement activation and leads to the effective lysis of target cells by forming a pore in the membrane, disrupting cellular integrity and function [25]. When the MAC is formed on membranes, a soluble counterpart sMAC, also referred to as the Terminal Complement Complex (TCC), can be formed. TCC levels are increased in various acute and chronic inflammatory diseases [25].

In view of the many ways by which complement activation can promote inflammation in various disease settings, we hypothesized that systemic inflammation in patients with cancer cachexia could be associated with complement activation. We set out to investigate the plasma levels of the initiating complement factors C1q and MBL and the effector complement factors C3a and TCC in blood samples from a cohort of pancreatic cancer patients with and without cachexia and inflammation. We report that cachectic patients with inflammation have higher systemic C3a and TCC concentrations than patients without inflammation or cachexia, and that these concentrations are strongly correlated overall. C3a and TCC concentrations also correlate with C1q levels, even though C1q concentrations do not differ between groups. These data may imply that the activation of the classical pathway of the complement system contributes to the inflammation seen in patients with cancer cachexia.

## 2. Materials and Methods

### 2.1. Patients

Patients undergoing pancreaticoduodenectomy at the Maastricht University Medical Centre (MUMC+) for suspected adenocarcinoma of the pancreas were enrolled in this study. Exclusion criteria included the use of systemic glucocorticoids in the past four weeks, neoadjuvant chemo- and/or radiotherapy, and the presence of another malignancy. This study was approved by the Medical Ethical Board of the Maastricht University Medical Center in line with the ethical guidelines of the 1975 Declaration of Helsinki, and written informed consent was obtained from each subject (METC 13-4-107 and 2019-0977).

### 2.2. Diagnosis of Cancer Cachexia and Screening of Cachexia Status

Cachexia was defined according to the international consensus definition as (1) weight loss >5% over the past 6 months in the absence of starvation, and/or (2) BMI < 20 kg/m^2^ and >2% ongoing weight loss, and/or (3) sarcopenia and >2% ongoing weight loss [3]. Patients were diagnosed with cancer cachexia if ≥1 of the criteria were met. The cachexia status of the patients was assessed by a trained physician in the outpatient clinic. The screening included measurements of body weight and height and patient-reported weight loss in the last six months.

To assess the presence of sarcopenia, body composition was investigated by analysis of computed tomography (CT) imaging scans and sliceOmatic 5.0 software (TomoVision, Magog, Canada). Skeletal muscle mass was quantified on a cross-sectional CT image at the third lumbar (L3) vertebra that was preoperatively acquired for diagnostic purposes. Using predefined Hounsfield unit (HU) ranges, the total cross-sectional area (cm^2^) of skeletal muscle tissue (−29 to 150 HU) was determined. In addition, the total areas of visceral adipose tissue (VAT, −150 to −50 HU) and subcutaneous adipose tissue (SAT), as well as intramuscular adipose tissue (IMAT) (−190 to −30 HU), were assessed. Tissue areas (cm^2^) were adjusted for height to calculate the respective L3 indices (L3-SMI, L3-VAT, L3-SAT) in cm^2^/m^2^, which correspond well with total body muscle and adipose tissue mass [26]. Previously published validated sex-specific cut-off values (SMI, 45.1 cm^2^/m^2^ for men and 36.9 cm^2^/m^2^ for women) that were established from a cohort including pancreatic cancer patients were used for the CT-derived body composition analysis [27]. In addition, the skeletal muscle radiation attenuation (M-RA) was calculated as the average HU value of the total tissue area for muscle (i.e., within the specified range of −29 to 150 HU).

Systemic inflammation was assessed by measuring plasma C-reactive protein (CRP) levels preoperatively (routine in-hospital laboratory test, MUMC+). CRP values were considered to be elevated when they exceeded 10 mg/L [28].

### 2.3. Collection of Plasma Samples

Prior to the start of surgery, after a minimum of eight hours of fasting, venous blood was collected in EDTA tubes and stored on ice until further processing. The blood was centrifuged at 1150× *g* at 4 °C for 12 min without brake. Plasma aliquots were stored at −80 °C until analysis.

### 2.4. ELISAs

Plasma C1q, MBL, C3a, TCC, and IL-6 concentrations were measured by enzyme-linked immunosorbent assay (ELISA) in accordance with the manufacturer’s instructions (Hycult Biotech, Uden, The Netherlands, for the complement factors; U-CyTech biosciences, Utrecht, The Netherlands, for IL-6). All plasma samples were analyzed in duplicate in the same run. The intra- and inter-assay coefficients of variance of the various assays were <10%. TCC concentrations were presented given in ‘milli arbitrary units’ (mAU)/mL.

### 2.5. Statistical Analysis

Statistical analysis was performed using Prism 7.0 for Windows (GraphPad Software Inc., San Diego, CA, USA), RStudio (v. 1.2.5033, Affero General Public License, Boston, MA, USA). Data are presented as median and interquartile range. Differences between groups were analyzed by the Kruskal–Wallis test followed by Dunn’s post testing. Correlations were calculated using Spearman’s correlation coefficient. *p*-values < 0.05 were considered statistically significant.

## 3. Results

### 3.1. Patient Characteristics

Sixty-two patients with pancreatic cancer were included in this study. They had a median age of 68.3 years and a median BMI of 24.4 kg/m^2^. In total, 35 males and 27 females were studied. The median weight loss percentage in the cohort was 7.2. CT scan-based body composition analysis showed that 45.2% of patients were sarcopenic, with a median L3-SMI of 46.7 cm^2^/m^2^ for males and of 36.1 cm^2^/m^2^ for females. The median M-RA value was 35.5 HU for the whole cohort. L3-VAT and L3-SAT indices were 63.9 and 43.6 cm^2^/m^2^ for males and 28.9 and 67.6 cm^2^/m^2^ for females, respectively. CRP concentrations were assessed as a measure of systemic inflammation, and found to have a median value of 9.0 mg/L. Additional descriptive patient characteristics can be found in Table 1.

Based on the consensus definition of cancer cachexia, we next subdivided the group into patients with and without cachexia. In line with the literature on the prevalence of cachexia in pancreatic cancer patients, we found that 49 patients (79%) were cachectic and 13 patients (21%) were not. Patients with cachexia had a significantly lower BMI as compared to patients without cachexia. Since cachexia is associated with inflammation, and because activation of the complement system is a key feature of many inflammatory conditions, we further subdivided the cachexia group into patients with and without inflammation, as evidenced by elevated systemic CRP levels. The median CRP concentration in the group with cachexia but without inflammation was 4.0 vs. 22.0 in the group with cachexia with inflammation. Body weight loss was similar between the groups with and without inflammation (median 9.5% IQR 7.3–13.9 vs. 9.7% IQR 5.7–12.7).

### 3.2. Concentrations of Complement Factors in the Blood of Patients with and without Weight Loss and Inflammation

We next assessed if we could detect activated complement factors with a key effector role in the blood of pancreatic cancer patients. Out of the 62 patients of the full cohort, we were able to assess levels of TCC, the ultimate result of activation of the complement system, for 61 patients. The median systemic concentration of TCC for the whole cohort was 2006 mAU/mL (IQR 1719–2603). Interestingly, TCC concentrations differed markedly among the subgroups (see Figure 1A), with the lowest values in the group without cachexia (1805 mAU/mL, IQR 1552–2569), intermediate values in the group with cachexia but without inflammation (1939 mAU/mL, IQR 1725–2311), and the highest values in the group with both cachexia and systemic inflammation (2298 mAU/mL, IQR 2022–3058, *p* = 0.0511).

A similar pattern was observed when we analyzed the concentrations of C3a, a split product of the central complement component C3 that indicates activation of the complement system, which we could analyze for 51 patients (see Figure 1B). C3a levels differed significantly across the groups (*p* = 0.0186), with higher values in the group with cachexia and inflammation (median 102.4 ng/mL, IQR 89.4–158.0) in comparison to the group with cachexia without inflammation (median 81.4 ng/mL, IQR 47.9–124.0) and the group without cachexia (median 61.6 ng/mL, IQR 46.8–86.8). Post testing revealed that the significance was driven by the difference between the non-cachectic group and the group with cachexia and inflammation (*p* = 0.0167), whereas the group without inflammation showed a *p*-value of 0.1720 vs. this latter group.

To gain more insight into initiating complement factors that could be responsible for complement activation in patients with cachexia and inflammation, we next assessed the systemic concentrations of C1q and MBL, the initiating factors of the classical and the lectin pathway, respectively. Whereas we could detect both C1q and MBL in the plasma of 60 patients, we did not observe any significant differences in their concentrations between the different groups. For C1q, the median concentrations were 10.2 µg/mL (IQR 9.4–12.6) vs. 10.3 µg/mL (IQR 8.3–13.3) vs. 11.3 µg/mL (IQR 7.3–13.7) in the groups without cachexia, with cachexia but without inflammation, and with cachexia and inflammation, respectively (*p* = 0.882, see Figure 1C). For MBL, the median concentrations were 694.4 ng/mL (IQR 385.8–1503.0) vs. 333.7 ng/mL (IQR 191.0–1185.8) vs. 406.4 ng/mL (IQR 138.2–1564.7) for these respective groups (*p* = 0.730, see Figure 1D).

### 3.3. Correlation Analysis of Complement Factors and Cachexia Parameters

Because the generation of TCC can only occur when the preceding complement factors in the complement cascade have been activated, we next investigated the association between concentrations of TCC and C3a, one of the upstream activated complement factors, in the blood across the entire cohort of patients. Importantly, we found a strong positive correlation between TCC and C3a concentrations (r_s_ = 0.4684, *p* = 0.0005, see Figure 2), supporting that patients with cachexia and inflammation display systemic complement activation. Additional correlation analyses revealed that C1q concentrations were also correlated to both TCC and C3a (r_s_ = 0.3002, *p* = 0.0188, and 0.3942, *p* = 0.0042, respectively, see Figure 2). MBL concentrations were not associated with any of the other complement factors analyzed in the Spearman correlation analyses.

To study potential interactions between complement activation and hallmarks of cachexia such as weight loss, body composition features (e.g., skeletal muscle index, adipose tissue indices, and skeletal muscle radiation attenuation), and BMI, we extended the correlation analyses (see Figure 3). Whereas BMI and body composition features showed the expected strong correlations, neither of these cachexia-related parameters was correlated to any of the complement factors assessed in this study.

## 4. Discussion

In the present study, we observed the systemic activation of the complement system in patients with pancreatic cancer who were cachectic and who displayed a systemic inflammatory response as evidenced by elevated CRP levels. Both C3a, a cleavage product of the central complement C3 component with potent pro-inflammatory functions, and TCC, an end product of complement activation, were present at higher levels in these patients as compared to those who were not cachectic, even though the latter group also displayed an elevation in CRP concentrations. The association between C1q and C3a, as well as TCC, in these patients, may indicate that the classical pathway of the complement system is involved in complement activation in patients with pancreatic cancer.

It is well known that inflammation plays a pivotal role in the development of cachexia in cancer patients [29,30]. As such, cachexia is generally associated with increased levels of inflammatory molecules such as CRP, TNF-α, IL-6, and IL-8. In the current study, we confirmed that cachectic patients with inflammation as defined by elevated CRP levels displayed higher circulating IL-6 concentrations. These cytokines promote tissue wasting in cachexia through the activation of transcription factors such as NF-kappa B [5,31]. Analogous to these pro-inflammatory cytokines, complement factors may be produced by cancer cells, immune cells, or metabolic target cells, such as hepatocytes or adipocytes [32,33,34,35].

In addition to their classical pro-inflammatory functions, various complement factors have been shown to have a tumor-promoting role in several types of cancer [36]. They may promote cellular proliferation, invasion, and migration, and mediate the development of an immunosuppressive microenvironment. In view of these contributions of complement to cancer progression, it is not surprising that we found increased complement activation products in the plasma of patients with cachexia and inflammation, which are associated with more advanced cancer [37].

An important next step will be to identify the cellular and/or tissue sources of complement activation, and their relative contributions to systemic active complement levels. One clue in this context may come from the association between the levels of C1q, the initiating factor of the classical complement pathway, and the levels of complement effectors C3a and TCC that we observed. C1q can instigate complement activation after direct binding to apoptotic cancer cells. It can also activate the complement cascade after binding to IgM or IgG antibody–antigen complexes, which may occur in pancreatic tumors as a result of neo-antigen expression by tumor cells. Both processes lead to the clearance of cells through macrophage phagocytosis, which could promote chronic inflammation as seen in cachexia.

Alternatively, the compromised gut barrier integrity that has been observed in models of cancer cachexia [38,39] and cachectic cancer patients [40] may promote complement activation through the classical pathway as a result of the formation of antibodies complexed to translocated bacteria. Furthermore, we have previously shown that both C1q, C3a, and MAC are deposited on steatotic hepatocytes in obese patients with nonalcoholic fatty liver disease [41]. Since it is known that cancer cachexia is also associated with hepatic steatosis, it should be investigated if lipid accumulation in the liver of cachectic patients leads to deposition of activated complement factors as well. In addition, older studies have shown that several complement factors, including C1, C2, C4, and C3, can be synthesized by skeletal muscle cells in vitro [42,43], and that they are upregulated by pro-inflammatory cytokines including IL-1 [42]. Proteomic and transcriptomic analyses support a key role for complement C3 in myogenesis [44].

Another important question that should be addressed concerns the potential effects of complement activation in the context of cancer cachexia. Given that C3a and TCC were specifically increased in those cachectic patients who displayed systemic inflammation, it is tempting to assume that complement activation in cachexia may propagate or even initiate the many pro-inflammatory events that play a central role in its pathogenesis. Furthermore, it is well known that the complement system can affect tissue homeostasis and regeneration in various conditions. For example, complement factors play a role in removing cellular debris during skeletal muscle remodeling after injury [45]. In particular, complement C3 has been shown to be essential for physiological tissue regeneration as evident from the inability of damaged cells to be cleared from C3-deficient mice and the subsequent fibrosis that develops [46,47]. At the same time, complement is known to participate in muscle destruction in inflammatory myopathies, including myasthenia gravis, dermatomyositis, and dysferlinopathies [48]. In view of the accumulating evidence for the impact of inflammatory events on the skeletal muscle microenvironment in the setting of cancer cachexia [49], complement may also contribute to muscle breakdown in cachexia. In line with this, it was recently found that complement is activated in the skeletal muscle of pancreatic cancer patients, and that C3-deficient mice displayed attenuated muscle atrophy in the KPC model of pancreatic cancer cachexia (Dr. AR Judge, University of Florida Health Science Center, personal communication). C1q has also been reported to reduce muscle regeneration in mice [50]. All in all, it remains to be established to what side the regenerative and destructive effects of complement in muscle are balanced in the context of cancer cachexia.

This balance is also dependent on the level of expression of regulatory molecules that inactivate complement factors at various steps of the complement cascade, which should be investigated in metabolic tissues of patients with cancer cachexia. It has been shown that myoblasts are protected from complement-mediated destruction and cell lysis by their abundant expression of the regulatory molecules CD46, CD59, and C4BP [51], but the expression of these complement inhibitory factors by mature myofibers in vivo is unknown. Similarly, human hepatocytes are characterized by high expression of a battery of complement inhibitors, including CD46, CD55, CD59, and factor H [52,53], although it is not known whether their expression is stable in metabolic disorders such as obesity or cachexia.

Besides the functions of complement factors in inflammation and tissue regeneration, it is noteworthy that adipocyte-derived C3a and its desarginated form, also known as acylation-stimulating protein, have been shown to affect lipid metabolism in adipocytes by stimulating triglyceride synthesis through the inhibition of hormone-sensitive lipase and by increasing glucose uptake [14]. In addition, systemic levels of complement C3 and C4 have been shown to be higher in patients with metabolic syndrome [54], and C1q expression has been reported to be dysregulated in adipocytes during insulin resistance [35], which also occurs in cachexia [55]. Complement factor D, also referred to as adipsin, is another complement protein produced by adipocytes that appears to affect metabolism by triggering C3a-dependent adipogenic signaling in adipose tissue as well as insulin secretion by pancreatic beta cells [14]. It would be interesting to investigate if the aberrations in lipid metabolism in cancer cachexia are associated with changes in these complement factors, even when the current consensus is that the expansion of adipose tissue, not the atrophy, leads to complement activation.

Even though we showed evidence for complement activation at multiple levels, some potential limitations of this study should be mentioned. First of all, the study population was relatively small, with a particularly minor fraction of patients displaying no cachexia, as is to be expected in pancreatic cancer. It will be important to expand this study to other cancer types with a lower cachexia prevalence to address this concern. In such a follow-up study, additional inflammation parameters to CRP, such as TNF-α, IFN-γ, and/or IL-1, should be included to gain more insight into the potential mechanistic links between complement activation and pro-inflammatory processes in patients with cancer cachexia. In particular, the analysis of other complement factors that have been not only implicated in inflammation but also in metabolism and regeneration, including C5a, properdin, factor D, and factor H, will be informative, next to the analysis of complement regulatory proteins, as discussed above. Such a study should keep the strengths of the current study design, with the analysis of cleavage products of complement factors reflecting complement activation, instead of detection of intact complement fragments (as is frequently seen in the literature), and a thorough characterization of cachexia according to the consensus definition [3]. Finally, it is important to note that the ‘no cachexia group’ had elevated systemic CRP levels, indicating inflammation. The underlying causes of inflammation-related CRP increases might vary for each patient. However, since we first stratified the patients according to their cachexia status before subdividing the cachexia group according to CRP level, we consider that CRP elevations in the ‘no cachexia’ group were not related to the metabolic and inflammatory alterations that come with cachexia. Our data support this notion since the higher CRP levels of the ‘no cachexia’ group did not translate into increases in other inflammatory molecules known to be elevated in cachexia, such as IL-6, or the complement factors C3a, TCC, and C1q.

## 5. Conclusions

In conclusion, systemic inflammation in patients with cancer cachexia is associated with the activation of key effector complement factors. The correlations between C1q and C3a/TCC suggest that the classical complement pathway could play a role in complement activation in patients with pancreatic cancer. Given that several complement-targeting therapeutics are already in clinical use [56], more evidence for complement activation as an important inflammatory process in cachexia could lead to intervention studies directed at complement inhibition to treat cachexia. However, first, more studies should be performed to confirm a role for complement activation in the pathogenesis of cancer cachexia.

## Figures and Tables

**Figure 1 cancers-13-05767-f001:**
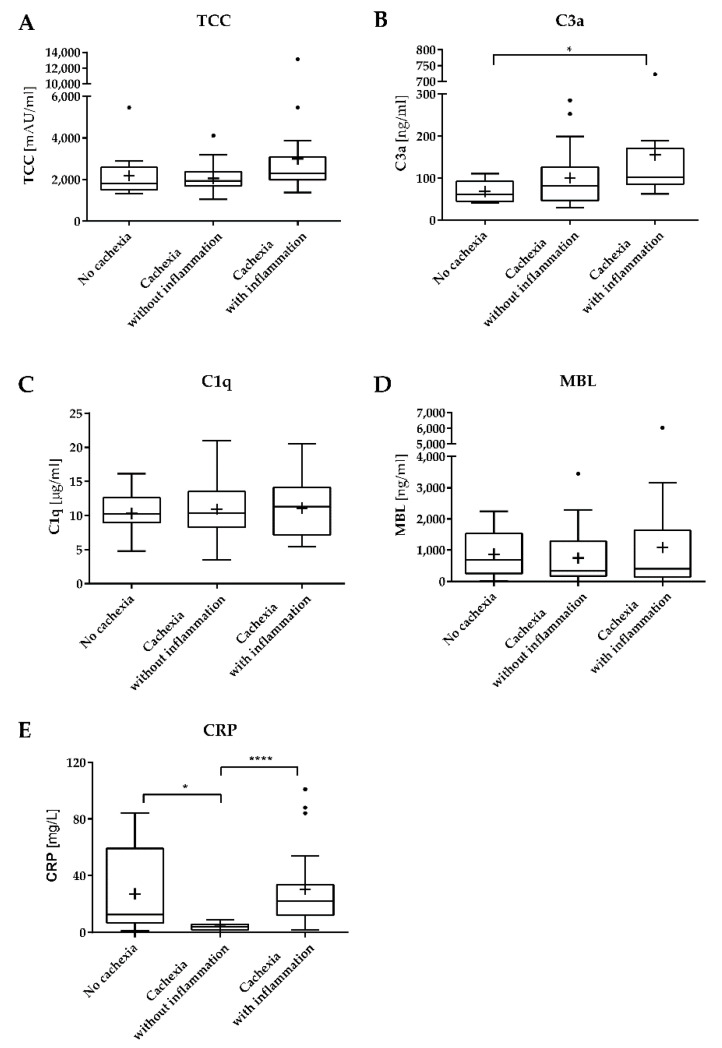
Plasma concentrations of C3a, TCC, C1q, MBL, and CRP in patients with pancreatic cancer. Tukey box-and-whisker plots are displayed, showing significantly elevated C3a concentrations in the blood of patients with cachexia and inflammation (panel **A**, *p* = 0.0186). TCC concentrations are higher in this group as well, with a trend towards significance (panel **B**, *p* = 0.0511). C1q (panel **C**) and MBL (panel **D**) concentrations do not differ between the groups. CRP concentrations were lower in cachectic patients without inflammation as compared to cachectic patients with inflammation or patients without cachexia (panel **E**, *p* < 0.0001, *p* = 0.0137, respectively). The line reflects the median; the hinges of the boxes are drawn at the 25th and 75th percentile. The dots reflect the outliers as defined by the Tukey method. The mean concentrations are indicated by the ‘plus’ sign. * *p* ≤ 0.05; **** *p* ≤ 0.0001.

**Figure 2 cancers-13-05767-f002:**
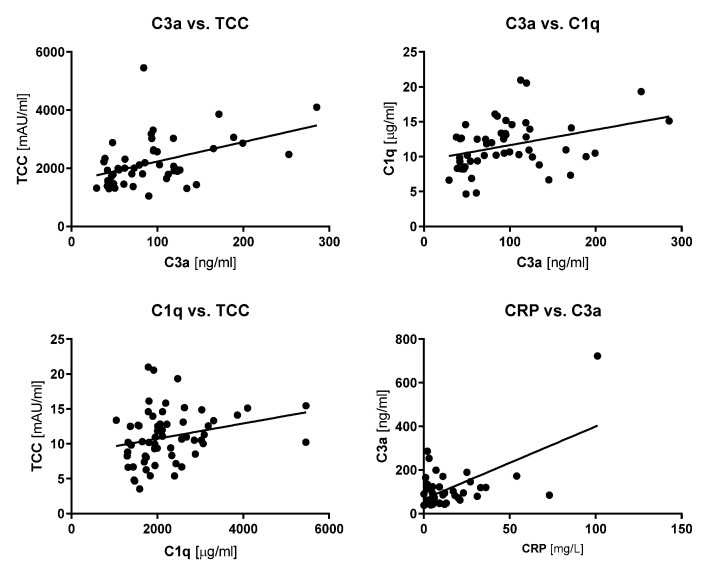
Associations between C3a, TCC, C1q, and CRP. x- and y-plots showing significant correlations between C3a and TCC as well as between C3a and C1q, and TCC and C1q, but no significant correlation between CRP and C3a.

**Figure 3 cancers-13-05767-f003:**
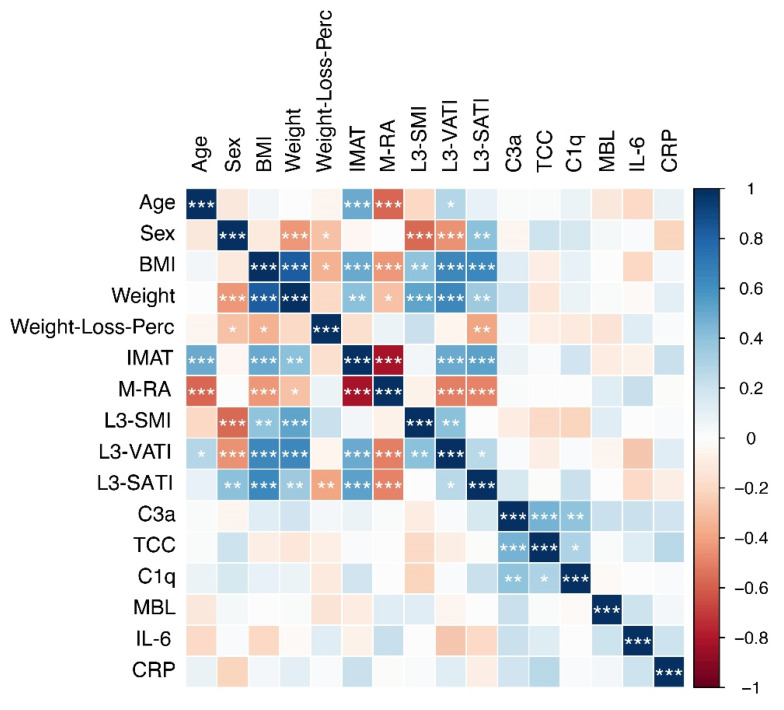
Correlation plot showing Spearman correlations between indicated cachexia-related variables and complement factors. Positive correlations are shown in blue, negative correlations in red. The color intensity indicates the strength of the correlation. The asterisks indicate the level of statistical significance (* *p* ≤ 0.05, ** *p* ≤ 0.01, *** *p* ≤ 0.001).

**Table 1 cancers-13-05767-t001:** General characteristics of patients with pancreatic tumor according to cachectic state.

Variable	Overall	No Cachexia	Cachexia	Cachexia	*p*
			Without Inflammation	With Inflammation	
*n*	62	13	26	23	-
Male/Female (%)	35/27 (56.5/43.5)	6/7 (46.2/53.8)	14/12 (53.8/46.2)	15/8 (65.2/34.8)	0.511
Age (years)	68.3 (63.1, 75.2)	67.6 (61.4, 76.3)	70.2 (63.2, 74.8)	67.8 (64.9, 74.2)	0.997
Weight (kg)	68.5 (58.7, 81.5)	76.5 (68.0, 82.2)	67.1 (56.5, 78.1)	67.0 (59.0, 82.3)	0.150
BMI (kg/m^2^)	24.4 (21.7, 26.8)	26.9 (25.6, 28.3)	23.5 (21.6, 26.1) ^†^	23.0 (20.7, 25.7) ^†^	0.015
Weight loss percentage (%)	7.2 (4,0, 12.2)	1.9 (0.0, 3.0)	9.7 (5.7, 12.7) ^†^	9.5 (7.3, 13.9) ^†^	<0.001
L3-IMAT (cm^2^)					
Male	14.6 (7.2, 19.5)	18.0 (10.3, 21.6)	12.2 (6.3, 19.7)	13.5 (7.4, 17.6)	0.581
Female	9.7 (7.6, 19.2)	13.9 (10.7, 18.4)	8.1 (7.7, 22.3)	7.7 (7.2, 13.6)	0.292
M-RA (HU)					
Male	35.3 (30.8, 42.8)	35.5 (33.7, 38.3)	32.5 (29.4, 41.2)	35.3 (32.5, 44.2)	0.757
Female	37.4 (30.0, 40.7)	37.4 (31.4, 40.0)	35.6 (28.5, 40.9)	36.8 (33.1, 41.6)	0.843
L3-SMI (cm^2^/m^2^)					
Male	46.7 (41.7, 50.3)	47.0 (45.6, 48.3)	47.6 (43.2, 51.3)	43.0 (40.3, 48.8)	0.564
Female	36.1 (34.2, 42.5)	38.9 (35.8, 42.7)	36.7 (34.9, 42.5)	35.1 (33.5, 39.4)	0.665
L3-VATI (cm^2^/m^2^)					
Male	63.9 (33.4, 79.7)	70.6 (66.3, 79.7)	64.6 (31.8, 85.1)	44.4 (30.5, 71.6)	0.221
Female	28.9 (12.4, 47.7)	39.7 (25.1, 48.8)	29.0 (10.2, 49.4)	28.7 (4.8, 37.3)	0.446
L3-SATI (cm^2^/m^2^)					
Male	43.6 (31.6, 57.3)	52.1 (39.8, 59.2)	47.1 (34.8, 56.0)	39.7 (27.1, 45.6)	0.349
Female	67.6 (49.7, 87.8)	87.1 (63.8, 88.8)	59.3 (46.7, 77.2)	63.1 (30.0, 85.3)	0.413
C3a (ng/mL)	90,0 (52.3, 120.8)	61.6 (46.8, 86.8)	81.4 (47.9, 124.0)	102.4 (89.4, 158.0) ^†^	0.019
TCC (mAU/mL)	2005.5 (1718.6, 2602.9)	1805.1 (1552.1, 2569.4)	1938.5 (1724.6, 2310.6)	2297.5 (2021.5, 3057.8)	0.051
C1q (μg/mL)	10.5 (8.2, 13.0)	10.2 (9.4, 12.6)	10.3 (8.3, 13.2)	11.3 (7.3, 13.7)	0.882
MBL (ng/mL)	437.3 (167.8, 1498.9)	694.4 (385.8, 1503.0)	333.7 (191.0, 1185.8)	406.4 (138.2, 1564.7)	0.730
IL-6 (pg/mL)	5.1 (2.9, 15.0)	4.4 (1.3, 9.8)	4.1 (1.9, 6.5)	11.2 (6.0, 31.7) ^‡^	0.042
CRP (mg/L)	9.0 (4.0, 20.8)	12.5 (10.3, 31.8)	4.0 (2.0, 5.3) ^†^	22.0 (12.8, 32.5) ^‡^	<0.001

The data are presented as median and interquartile range. Groups were compared using the Kruskal–Wallis test. ^†^ Significant difference in comparison to the no cachexia group. ^‡^ Significant difference in comparison to the cachexia without inflammation group. BMI: body mass index; HU: Hounsfield unit; L3_IMAT: L3-intermuscular adipose tissue; M-RA: muscle radiation attenuation; L3-SMI: L3-muscle index; L3-VATI: L3-visceral adipose tissue index; L3-SATI: L3-subcutaneous adipose tissue index; TCC: terminal complement complex; C1q: complement component 1q; MBL: mannose binding lectin; IL-6: interleukin 6; CRP: C-reactive protein. mAU/mL: milli arbitrary units/mL.

## Data Availability

The data presented in this study are available on request from the corresponding author.

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
