# Peer review of "Activation of the Complement System in Patients with Cancer Cachexia"

_cancers, 2021, doi:10.3390/cancers13225767_

Round 1

Reviewer 1 Report

This manuscript is well written, organized and presented. It describes how CRP-involved complement activation and assumed inflammation may play a role in disease progression in very selective populations of patients with pancreatic cancer who experience weight loss. 

While the information presented and developed are reasonable, strong arguments can be made that there are other possible things ongoing in this selected patient group that could influence the premise made and the conclusions reached in tis submission. Some of such concepts are presented below.  As addressing each comment included would significantly affect the tone and direction of the submitted work, it is deemed by this reviewer as unfair to ask such things as a prelude to recommending acceptance.  Most significantly, however, the presented work should be viewed as the start of an idea with some data to support the premises made.  Any suggestion that the data presented would support use of anti-complement system therapeutics as a treatment option in cachexic pancreatic patients is very premature and should be carefully worded so to not suggest this as a possible treatment, but to encourage more studies.

Some specific comments/questions:

When the specific patient populations in this study were described, an initial thought involved thinking about metabolism before considering any inflammatory pathways.  What is going on with the biochemical fuels needed to generate ATP energy (first using glucose, then fats, then proteins as fuel sources).  As the patients in this study have pancreatic cancer, one must wonder what is going on with insulin, glucagon, and digestive enzyme available for core metabolism needs.   If glucose metabolism is severely affected, one would expect there to be fat depletion (which would accompany some weight loss), and if fat stores are used up, protein will be degraded to provide amino acids that can be used to generate fuel (hence, possible muscle wasting).   Removing molecules from adipose and especially muscle tissues would cause some level of tissue damage which would be the main initiator of an inflammatory response.  This could/would involve CRP and complement protein pathways to help inflammatory pathways to clean up the damaged tissues and start tissue repair processes.

Since these are pancreatic cancer patients:

  • What is their insulin synthesis/secretion values?
  • What is their blood glucose/A1c level and how does this change during their illness?
  • If use of glucose as a fuel is compromised, one immediately thinks of utilizing fat stores to produce energy (hence the “fat depletion” monitored.) Once fats are depleted, proteins will be catabolized into intermediates to enter ATP energy generating pathways.  An early step in amino acid catabolism involves transamination and deamination reactions.  Were there changes in AST/ALT levels, and/or changes in blood urea nitrogen (BUN) correlate with muscle wasting?
  • How are blood total protein and albumin levels correlate? One would expect blood albumin to be used during cachexia as a source of amino acids to produce energy.  If these levels are depleted, one might expect muscle proteins will next be broken down.  What happened to blood albumin levels in this patient group?
  • One assumes skeletal muscles will first be broken down – but were there any signs of cardiac muscle breakdown especially in the most severely diseased individuals? What happens to Cardiac markers in blood correlate with disease progression?

  1. In light of the arguments that CRP is useful marker to support their arguments, how do the authors explain that the patient group “without Cachexia” has a CRP mean of 12.5 (which is above their cut off for “inflammation” (i.e., > 10 ug/ml). If CRP is above the defined “inflammation level,” does it similarly affect complement protein markers in this selected group?
  2. What exactly are the mAU/ml values presented when discussing the terminal attack complex measurements? This abbreviation needs to be more definitively and directly described in the Methods section, in the Tables where it is included, and in a Abbreviations paragraph.
  3. Concerning the Membrane Attack Complex (TCC), CRP does not always result in TCC formation. Some CRP effects only go so far as to deposit C3b, so to opsonize the target. Hence, one might not expect a correlation between CRP and TCC.
  4. Also concerning TCC, it is a “membrane” complex.  If it is found in plasma, it must have been sloughed off the cell membrane where it formed.  Plasma TCC would be expected to have lipid (macrovesicles) associated with it.  Was this involved in how the samples were collected and analyzed?
  5. Is the mannose binding protein the same thing, or related to the Serum Amyloid P component (SAP) - which is pentraxin with sequence homology to the CRP pentraxin?
  6. Is the C1q measured in plasma a free C1q molecule or is it found as a component of intact C1 (i.e., C1qC1r2C1s2). Is C1 dissociated in chelated plasma?  Is the measurement of C1q in the assay used dependent on whether C1q is free or complexed?
  7. Do IL-6 levels in plasma correlate with CRP levels (since IL6 is the primary cytokine needed to signal hepatic production of CRP).
  8. How do the authors explain that, if there is muscle wasting (i.e., if muscle proteins are being broken down to generate amino acids to be used as fuel for ATP producing pathways), why would the body use these precious amino acids to “synthesize” CRP and complement proteins and promote inflammation?
  9. The authors mention that plasma samples were collected “prior to surgery” but do not describe what or why the surgery (or surgeries) were done. Please elaborate.
  10. The extent of tissue damaging disease occurring when samples were collected is a factor that is important in understanding CRP levels and ongoing inflammation.  Where single samples collected for each of the patient included in this study?  If some patients had multiple samples collected over time, how are all the parameters discussed affected?

Author Response

Please see the attachment for our point-by-point response to your comments.

Reviewer 2 Report

The manuscript by Min Deng et al., entitled “Activation of the Complement System in Patients with Cancer Cachexia,” is well-written and scientifically sounds. Given the profound gap in knowledge, this is an important article suitable for Cancers journal.

Major suggestions that need to be addressed:

  1. CRP-specific suggestions:
    1. Add CRP plasma levels in three groups to Fig. 1
    2. Add CRP/C3a association plot to Fig. 2
    3. Line 268: I suggest bringing the reader’s attention here to the difference in statistics (median, IQR) in No cachexia Cachexia with Inflammation groups. C3a is not elevated in the No cachexia group despite elevation in CRP.
    4. Line 270: to make this conclusion, you need to show no statistically significant difference (SSD) between CRP in No cachexia Cachexia with Inflammation groups. Comment on the difference in skewness in the two samples. IF there is an observable SSD, I would suggest changing emphasis to extend in the inflammation rather than associating it with cachexia with inflammation.
      1. Line 71: The classical pathway of complement activation is initiated by binding of the C1q molecule to … CRP. An increase in CRP was shown to be associated with later stages of PDAC. That is, there is a bit of circular logic that CRP was used to define the inflammation group, but is also known to activate the complement system. So finding increased complement activation in this group is not surprising. This is still the first demonstration of complement activation in humans with cachexia, but it needs to be acknowledged that there are many ways to define inflammation, and it is not clear that all of them would show a statistically significant association with complement activation.
    5. Statistical analysis:
      1. I suggest using the post-hoc Dunn test for your Kruskal–Wallis analysis to identify the exact groups with continuous data that have a significant difference. Your sample data sets do not follow normal distributions (i.e., CRR); you need to acknowledge this observation.  

Minor concerns that should be addressed:

  1. I would suggest addressing the following minor grammatical errors that could distract the reader:
    1. Line 30: mannose binding lectin > mannose-binding lectin
    2. Line 42: the survival
    3. Lines 126 and 139: inconsistent use of pre-operative/preoperative
    4. Line 147: a minimum of eight hours fasting > a minimum of eight hours of fasting
    5. Line 161: Kruskal Wallis-test to Kruskal-Wallis test
    6. Line 212: non-cachechtic group to non-cachectic group
    7. Line 251: This sentence may be unclear or hard to follow for the reader.
  2. Line 133: mention the cut-off values (cm2/m2) defining cachexia based on your calculation’s method
  3. Line 140: mention reference range for CRP.
  4. I would suggest combining Diagnosis of cancer cachexia with Screening of cachexia status. The first paragraph of the Screening immediately raises the question about the diagnosis criteria used by the physician.
  5. Markers:
    1. I suggest using additional markers to define inflammation, such as neutrophil to lymphocyte ratio (NLR), albumin to globulin ratio (ARG), or CRP/ALB ratio.
    2. If it is not plausible to do, justify the choice of CRP in comparison to other markers. Are they mutually exclusive?
  6. No cachexia group has a CRP of 12.5 [10.3, 31.8]. The group seems to have inflammation based on your inclusion criteria. I would suggest changing the group’s name to No cachexia with inflammation or adding a description/justification of increased CRP in this group. This dataset has a positively skewed distribution. Ideally, you may want to subdivide this category to No cachexia with inflammation and No cachexia without inflammation and comment on this analysis.
  7. Address increase in IL-6 in Cachexia with inflammation in Discussion

Author Response

(The authors gave the same response as above.)

Reviewer 3 Report

Deng et al., studied activation of the complement system in patients with cancer cachexia. In this study, author have presented in detailed background, required methodology and descriptive results and discussion. Minor comments to wards translational research, Metabolomics is a new methodology for cancer biomarker research area. metabolities are terminal point of the product of biological processes, therefore reflecting up/down stream targets such as  genetic, proteomic alterations, and environmental influences,  with having approximately 25 patients with or without inflammation data, and serum samples can do OMICs approach to fulfill this study towards new therapeutic approach.

Author Response

(The authors gave the same response as above.)
